# HeteroSFL: Split Federated Learning with heterogeneous clients and non-IID data

## Abstract

Split federated learning (SFL) is an emerging privacy-preserving decentralized learning scheme, which splits a machine learning model so that most of the computations are offloaded to the server. While SFL is edge-friendly, it has high communication cost and so existing SFL schemes focus on reducing the communication cost of homogeneous client-based systems. However a more realistic scenario is when clients are heterogeneous, i.e. they have different system capabilities including computing power and communication data rates. We focus on the heterogeneity due to different data rates since in SFL the computation in the client-end is quite small. In this paper, we propose HeteroSFL, the first SFL framework with heterogeneous clients that handles non-IID data with label distribution skew across clients and across groups of clients. HeteroSFL compresses data with different compression factors in low-end and high-end group using narrow and wide bottleneck layers (BL), respectively. It provides a mechanism to address the challenge of aggregating different-sized BL models, and utilizes bidirectional knowledge sharing (BDKS) to address the overfitting caused by the different label distributions across high- and low-end groups. We show that HeteroSFL achieves significant training time reduction with minimum accuracy loss compared to competing methods. Specifically, it can reduce the training time of SFL by $16\times$ to $256\times$ with $1.24\%$ to $5.59\%$ accuracy loss for VGG11 on CIFAR10 for non-IID data.

## 1 Introduction

Data security has become a big concern in training of Deep Neural Network (DNN)s, where typically raw data at edge is collected and processed by a central server. Federated learning (FL) McMahan et al. (2017) is a popular scheme aimed at preserving privacy in decentralized learning without sharing private data with the central server. The classical FL scheme trains the entire model on the client side, which is not practical for edge clients with limited computation and memory resources. Training an epoch of VGG11 model on an Nvidia Jetson board takes four hours, which is hundreds of times slower than on a GPU server Zhou et al. (2021). To address this drawback, split federated learning (SFL) Thapa et al. (2022) is proposed, where the model is split into a small front-end client-side model and a big back-end server-side model, and the client only needs to process and update the client-side model. The detailed training process is shown in Fig. 1(a). By splitting VGG11 after the third convolution layer, the number of computations at the client side is reduced by $10\times$. Thus, SFL facilitates resource-constrained clients to participate in training NN models.

One drawback of SFL is the high communication cost due to the back-and-forth transmission of activations and gradients of every sample in every epoch. In a realistic scenario, the clients in SFL have different system capabilities. For instance, a low-end client can have limited resources and only support low communication rate protocols, such as BLE and Zigbee, while a high-end client can support communication protocols with high data rates, such as WiFi and LTE. In this paper, we cluster the clients that support similar data rates into a group. In SFL, the server cannot start processing until it receives activations from *all* participating clients. Therefore, the server has to wait for data from the low-end group, which increases the overall training time significantly. Since the communication time of SFL is dominated by the low-end group, reducing the data size transmitted/received by low-end group can help reduce the training time.

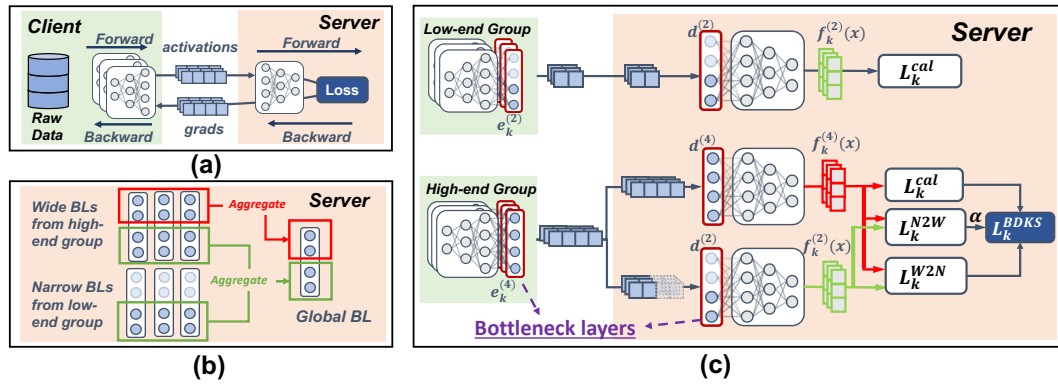

Figure 1: (a) Training of Split Federated Learning. The model is split into a small client-side model and a large server-side model. $K$ clients compute in parallel and send their activations to the server. The server computes the forward and backward process on the server-side model and sends the gradients back to clients. At the end of every epoch, the client-side models are aggregated by the server and sent back to the clients. (b) The parameters of the narrow BLs in low-end group of clients are aggregated with the subnetwork of wide BLs in high-end group of clients in HetBL. (c) Proposed HeteroSFL where the data of different clients are compressed by different ratios through use of different-sized BLs. To mitigate the over-fitting due to non-IID data, the loss function is replaced with logit calibration ($\mathcal{L}_k^{cal}$) and bi-directional knowledge sharing ($\mathcal{L}_k^{BDKS}$).

The traditional data compression methods like quantization Wang et al. (2022) or Top-k sparsity Liu et al. (2018) result in a significant accuracy drop. For example, Wang et al. (2022) reduces transmitted/received data size by $2\times$ at the cost of 5% accuracy loss. Recently, end-to-end trained bottleneck layer (BL) method proposed in Eshratifar et al. (2019); Ayad et al. (2021) reduces the data size while maintaining good accuracy performance. It achieves this by adding a convolution layer with fewer output channels (encoder) at the client-side and the corresponding mirrored structure (decoder) at the server side. Existing BL methods implement identical-sized BLs for all clients. This means that the communication data of both high-end and low-end groups is compressed by the same amount, resulting in the clients in the low-end group (with lower data rate) to take longer to complete and thus increase the overall training time. To reduce the training time, we propose that the low-end groups compress the data by a large ratio by using narrow BLs (designed using fewer channels in the convolution layer). In contrast, high-end groups can use wider BLs (designed using more channels) to maintain accuracy performance.

However, the different BL structures used in different groups cause two challenges. First, the parameters of the different-sized BL models cannot be directly aggregated using averaging due to the differences in the BL structures. The second challenge is the label distribution skew across clients and across groups of clients. Inter-client label distribution skew, where the number of samples per class differs from client to client Kairouz et al. (2021), has been well studied in FL Karimireddy et al. (2020); Tang et al. (2022); Zhang et al. (2022), and can be extended to alleviate the inter-client label skew in SFL to some extent. A recent study in FL recommendation system Maeng et al. (2022) highlighted the label skew across groups with different system capabilities, referred to as system-induced data heterogeneity. When groups with different system capabilities train models with different structures, the inter-group label skew makes the model over-fit to the data within their respective groups, leading to low accuracy. However, prior works on heterogeneous FL systems Diao et al. (2021); Horvath et al. (2021); Ilhan et al. (2023) neglect the inter-group label skew phenomenon when discussing inter-client label skew.

In this paper, we propose ***HeteroSFL***, the first SFL framework for heterogeneous clients that can tackle inter-client and inter-group label distribution skew with significant training time reduction and minimum accuracy loss. The training time reduction is achieved by compressing the transmitted data in high- and low-end groups differently through use of different-sized BLs, as shown in Fig.1(c). To address the aggregation problem of different-sized BLs, we design HetBL where narrow BLs are the subnetworks of wide BLs. And to address the inter-group label skew problem, we propose bi-directional knowledge sharing (BDKS), where the wide BL model learns knowledge on

underrepresented classes from the narrow BL model which is trained by the entire data set, and the narrow BL model learns from the wide BL model to avoid gradient interference between wide and narrow BL models. We also utilize logit calibration Zhang et al. (2022) to mitigate the inter-client label skew. Our extensive experiments show that HeteroSFL can reduce training time of SFL by $16\times$ to $256\times$ with only $1.24\%$ to $5.59\%$ accuracy loss for VGG11 on CIFAR10 for non-IID data. For the same training time reduction, HeteroSFL achieves accuracy improvement over other schemes; the improvement is $30.07\%$ over SampleReduce and $7.16\%$ over Top-k sparsity Liu et al. (2018).

## 2 RELATED WORK

**Split Federated Learning**  SFL Thapa et al. (2022) is a parallel version of Split Learning that was first proposed in Gupta & Raskar (2018). It splits the neural network model into a small client-side model and a big server-side model, and thereby offloads most of the computations to the server, as shown in Fig. 1(a). Similar to FL, SFL consists of two steps: local training and aggregation. In local training, the client-side model of client $k$ is trained by its local data $\boldsymbol{D}_k$ as follows:

$$\min_{f^s, f_k^c} \sum_{\{x,y\} \in \boldsymbol{D}_k} \mathcal{L}_k(f^s(f_k^c(x)), y) \tag{1}$$

where $\mathcal{L}(\cdot, \cdot)$ is the loss function and $f^s(\cdot)$ and $f_k^c(\cdot)$ represents the server-side model and client-side model. For convenience, we use $f_k(\cdot)$ to represent $f^s(f_k^c(\cdot))$ in the rest of the paper. At the end of every epoch of local training, the client-side models from all clients are aggregated on the server side by averaging and the global client-side model is broadcast back to the clients to initialize the client-side model for the next epoch.

**Data Compression Method in SFL**  In SFL, the transmission of activations and gradients for every sample in every epoch results in a high communication overhead. Recent developments in SFL include reducing the communication overhead by totally eliminating the gradient transmission He et al. (2020); Han et al. (2021), and reducing the number of training epochs Chen et al. (2021); Ayad et al. (2021), which cause significant accuracy loss. Existing works to compress the activations/gradients of DNN using JPEG Eshratifar et al. (2019), quantization Chen et al. (2021); Wang et al. (2022) and sparsity Liu et al. (2018) all suffer from high accuracy loss. End-to-end trainable BL, proposed in Eshratifar et al. (2019); Ayad et al. (2021) to compress the data size using trainable convolution layers, achieve good accuracy with large data compression. We build upon this promising technique to reduce the transmitted/received data size.

**FL with heterogeneous clients**  In FL, computation overhead is a major concern and the low-end group usually take longer, increasing the overall training time. Some works reduce the computation time of low-end group in FL through asynchronous training Recht et al. (2011), reducing the number of local training epochs Li et al. (2020) and reducing the number of processed samples Nishio & Yonetani (2019); Shin et al. (2022). The sample reduction method can be extended to SFL to reduce communication time but suffers from accuracy loss, as shown in Section 5. In Diao et al. (2021); Horvath et al. (2021); Zhu et al. (2022); Makhija et al. (2022); Lin et al. (2020), the low-end group only train a smaller network compared to the high-end group to reduce the computation time in FL. Since the client-side computation in SFL is quite small, we do not consider it. Instead, we focus on reducing the communication cost by compressing the transmitted/received data using BL.

**Label distribution skew**  The data processed by different clients usually suffers from inter-client label distribution skew, where the number of samples per class differs from client to client Kairouz et al. (2021), hampering convergence as presented in Karimireddy et al. (2020); Li et al. (2022c); Tang et al. (2022). In SFL and FL, the inter-client label skew causes the client-side model to overfit the local data. In SFL, this is addressed by increasing the frequency of synchronization at the cost of more communication overhead Li et al. (2022a) and training the client-side model using a more evenly distributed subset of the original dataset Cai & Wei (2022). In FL, some approaches address the label skew problem by restricting local updates to the global model Li et al. (2021; 2020); Karimireddy et al. (2020); Gao et al. (2022); others attempt to rectify the data distribution among clients using out-of-distribution datasets Tang et al. (2022); Xu et al. (2022) or using synthesized data generators Zhu et al. (2021); Chen et al. (2022). However, all these methods incur significant computation and communication overhead on the resource-limited clients and are unacceptable for SFL. In contrast, Zhang et al. (2022) adjusts logit values of minority classes to address the non-IID

issue without additional overhead. Thus we extend the use of logit calibration Zhang et al. (2022) to SFL to mitigate the effect of inter-client label skew in HeteroSFL. Note that none of the methods consider the problem of inter-group label skew.

## 3 BACKGROUND AND PROBLEM DESCRIPTION

In this paper, we focus on training-time reduction with minimum accuracy loss in SFL with heterogeneous clients processing data with inter-client and inter-group label skew. We assume the average data rate of every client is known beforehand and does not change during training. The overall training time of SFL could be calculated by:

$$T = \max_{k \in \boldsymbol{K}}((comm_k + comp_k)) \times n\_batches \times n\_epochs \tag{2}$$

$$comm_k = \frac{2 \times |f_k^c(x_k)| \times batch\_size}{r_k} \tag{3}$$

$comm_k$ and $comp_k$ are the communication and computation time of client $k$ to process a batch of samples. $comm_k$ is a function of the data size transmitted/received per sample and $r_k$, the data rate of client $k$. The data per sample, includes the client-to-server activations and server-to-client gradients, and its size given by $2 \times |f_k^c(x_k)|$. We refer to this data as communication data.

**Training time** In SFL, the computation time is negligible compared to the communication time. For example, in an SFL system training VGG11 using CIFAR10, when the client-side model includes the first 3 convolution layers, the communication data for a batch with 50 samples is around 52Mb, which takes around 1*min* to transmit using BLE while the computation time is 2 *sec* using the Jetson board. The training time is determined by the maximum time to process one batch of samples across clients since the server cannot start forwarding before receiving activations from all clients. Therefore, the overall training time is determined by the communication time of the low-end group in heterogeneous SFL. Due to the long communication time of low-end group, the data communicated by the low-end group should be compressed by larger ratio by using narrow BLs. On the other hand, the data communicated by the high-end group can be compressed by smaller ratio by using wide BLs to maintain the accuracy.

**Bottleneck layers.** Compressing $|f_k^c(x_k)|$ of low-end group reduces the communication time hence reducing the overall training time (See Eq.3). The bottleneck layer technique Eshratifar et al. (2019); Ayad et al. (2021) introduces two convolution layers, one at the end of client-side model to compress the channel size of activations from $\bar{p}$ to $p$, and the other one at the beginning of server-side model to decompress the activation from $p$ back to $\bar{p}$. Note that the number of channels of gradients is also compressed since it has the same size as that of activations. The BL is trained together with the whole network in an end-to-end manner and the extended network is used during inference. We denote the BL generating activations of $p$ channels as $p$-channel BL. Larger $p$ represents smaller compression ratio and thus higher accuracy performance since more information is sent to the server.

**Inter-group label distribution skew.** The local data in every client has different number of samples from different classes. While the existing works can mitigate the overfitting of client-side models caused by the inter-client label skew, they cannot address the inter-group label distribution skew. Fig.2(a) shows an example of the inter-group label skew with 10 classes, where the label distribution across high-, mid- and low-end groups is quite different due to system-induced data heterogeneity. As a result, models trained by different groups of clients tend to overfit to the label distribution of its own group, as shown in Section 4.2.

## 4 HETEROSFL

In this section, we propose Heterogeneous Split Federated Learning (*HeteroSFL*) where clients are grouped based on their data rates. The proposed HeteroSFL method shown in Fig.1(c) has two components: HetBL to address the different compression ratios of different groups through use of different-sized BL (Section 4.1), and bi-directional knowledge sharing (Section 4.2) to alleviate the overfitting caused by inter-group label skew. Note that while HeteroSFL can support multiple groups of clients, we describe HeteroSFL with only high-end and low-end groups for ease of understanding. We include results for multiple groups in Section 5.2.

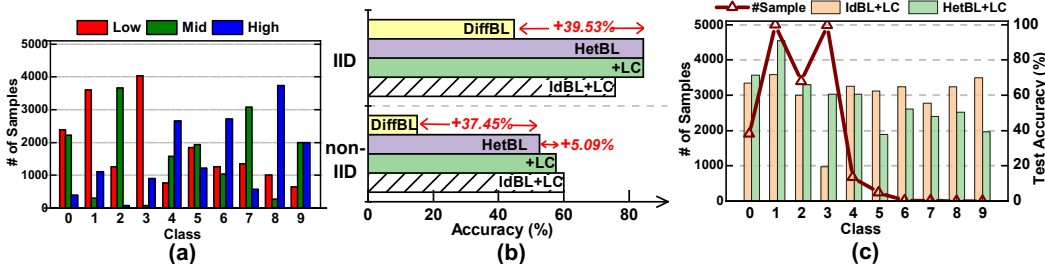

Figure 2: (a) An example of inter-group label skew caused by system-induced data heterogeneity. There are three groups of clients and the number of classes is 10.
(b) Accuracy performance of identical-size BL (IdBL) and variants of different-sized BLs with 40% high-end group. For different-sized BLs, the high-end group uses 16-channel BL and the low-end group uses 1-channel BL, while both groups use 1-channel BL in IdBL. The accuracy of HetBL is worse than IdBL even with mitigated inter-client label skew using logit calibration method (LC) Zhang et al. (2022) due to the inter-group label skew. (c) Class-wise number of samples in high-end group and class-wise accuracy of IdBL and HetBL for non-IID data. The accuracy of HetBL for underrepresented classes in high-end group is significantly lower than that of IdBL due to the inter-group label skew.

## 4.1 HETBL

Since the amount of data to be compressed by the high-end and low-end groups of clients is different, we propose to use different-sized BLs for different groups. Specifically, we use wide BL for high-end group and narrow BL for low-end group. Once the network is trained, the wide BL model is used for inference by all clients to achieve higher accuracy. Since the data size transmitted during inference is much smaller than in training, this is an acceptable compromise.

We define $\boldsymbol{e}_k^p \in \mathbb{R}^{\bar{p} \times p \times w \times w}$ as the encoder convolution layer of $p$-channel BL and $\boldsymbol{d}^p \in \mathbb{R}^{p \times \bar{p} \times w \times w}$ as the decoder convolution layer of the server. The wide BLs are trained only by the high-end group, and narrow BLs are trained only by low-end group. Due to the different structures, the parameters of wide and narrow BLs cannot be easily aggregated. In order to be able to aggregate parameters of wide and narrow BLs, we derive the narrow $p_N$-channel BL as a subnetwork of wide $p_W$-channel BL. Fig.1(b) shows the aggregation of HetBL where the first $p_N$ channels of wide BLs in high-end group are aggregated with the narrow BLs in low-end group, and the remaining $(p_W - p_N)$ channels of wide BLs are aggregated among the high-end group. Thus,

$$e^{p_W}[:, : p_N, :, :] = e^{p_N} = \sum_{k \in K_{low}} \gamma_k \cdot \boldsymbol{e}_k^{p_N} + \sum_{k \in K_{high}} \gamma_k \cdot \boldsymbol{e}_k^{p_W}[:, : p_N, :, :] \tag{4}$$

$$e^{p_W}[:, p_N : p_W, :, :] = \sum_{k \in K_{high}} \gamma_k \cdot \boldsymbol{e}_k^{p_W}[:, p_N : p_W, :, :] \tag{5}$$

where $e^{p_N}$ and $e^{p_W}$ are the global $p_N$-channel and $p_W$-channel BL after aggregation, respectively. $K_{low}$ and $K_{high}$ correspond to low-end and high-end group, and $\gamma_k$ is a factor proportional to the number of samples in clients.

Fig.2(b) shows the accuracy improvement of HetBL over DiffBL, a naive different-sized BL method that does not aggregate the narrow and wide BL models. HetBL increases the accuracy by 39.53% and 37.45% for IID and non-IID cases, respectively, highlighting the importance of aggregating narrow and wide BL models.

## 4.2 BI-DIRECTIONAL KNOWLEDGE SHARING

**Inter-group label skew cannot be addressed by inter-client label skew methods.** When the low-end and high-end group use different-sized BLs as in HetBL, we expect its accuracy to be better compared to identical-sized BL (IdBL), since the high-end group does not compress the data as much. Fig.2(b) shows the accuracy performance of IdBL and HetBL with 40% of clients in high-end group. HetBL outperforms IdBL for IID data. For non-IID data with inter-client label skew,

we add logit calibration (LC) Zhang et al. (2022) on top of HetBL. While HetBL+LC improves the accuracy of non-IID case by $5.09\%$, it is still worse than IdBL+LC. To better understand the cause for poor performance, Fig.2(c) shows the class-wise accuracy comparison of the two methods as well as the number of samples per class processed by the high-end group. We see that while the accuracy of classes $0$ to $3$ using HetBL is higher than IdBL, the accuracy drops in the underrepresented classes with insufficient number of samples. Therefore, the wide BL model over-fits the underrepresented classes $4$ to $9$ during SFL training.

To better understand the cause for over-fitting, consider Eq.4 and Eq.5 where only the first several channels of wide BL are trained by both groups, and the remaining channels of wide BL are only trained by high-end group. With inter-group label skew, the number of samples of some classes is insufficient as shown in Fig.2(a). Training wide BL model with data that includes underrepresented classes results in lower accuracy, as shown in Fig.2(c).

We define $\mathcal{R}$ to be the set of underrepresented classes processed by high-end group where the number of samples per class is less than $\theta\%$ of its share in the global dataset, and $\mathcal{J}$ the remaining set of classes (with sufficient number of samples). Thus $\mathcal{R}$ is a subset of minority classes in high-end group. To avoid over-fitting caused by underrepresented classes, the wide BL should learn about $\mathcal{R}$ classes from other models that have been trained using sufficient number of samples. The narrow BL model is a subnetwork of the wide BL, and thus is able to be trained by the data both from low-end and high-end groups. Therefore, we use the narrow BL model to teach the wide BL model.

**Narrow-to-Wide (N2W) Knowledge Sharing** The wide BL model learns from the narrow BL model using logits of $\mathcal{R}$ classes generated by narrow BL model, as shown below:

$$\mathcal{L}_k^{N2W}(f_k^{p_W}(x), f_k^{p_N}(x)) = -\sum_{i \in \mathcal{R}}(\mathcal{P}^{p_N}(x) \cdot log\mathcal{P}^{p_W}(x)) - (\sum_{i \in \mathcal{J}}\mathcal{P}_i^{p_N}(x)) \cdot log(\sum_{i \in \mathcal{J}}\mathcal{P}_i^{p_W}(x)) \quad (6)$$

where $\mathcal{P}_i^p(x)$ is the softmax probability of class $i$ using p-channel BLs. The first term helps the wide BL model to learn about underrepresented $\mathcal{R}$ classes from the narrow BL model and the second term ensures that the wide BL model does not learn about $\mathcal{J}$ classes from the narrow BL model.

**Wide-to-Narrow (W2N) Knowledge Sharing** We observe that the gradients of wide and narrow BL models interfere with each other when training with one-hot label, leading to an accuracy drop, a phenomenon that is also reported in Horvath et al. (2021); Mohtashami et al. (2022); Baek et al. (2022). To eliminate the interference, we train the narrow BL models in high-end group by distilling knowledge from wide BL model. The loss function of training narrow BL model in high-end group is as follows:

$$\mathcal{L}_k^{W2N}(f_k^{p_N}(x), f_k^{p_W}(x)) = CE(f_k^{p_N}(x), f_k^{p_W}(x)) \quad (7)$$

where $CE(\cdot, \cdot)$ is the softmax cross-entropy, and $f_k^{p_N}(x)$ and $f_k^{p_W}(x)$ are the logits generated by narrow BL and wide BL, respectively. Since the narrow BL is a subnetwork of wide BL, training narrow BL individually does not introduce additional communication and computation overhead on the client side, but doubles the computation overhead at the server end.

**Bi-directional Knowledge Sharing** We propose the BDKS loss function which combines the W2N and N2W knowledge for the high-end group. It consists of three elements: logit calibration and N2W knowledge sharing to train wide BL, and W2N knowledge sharing to train narrow BL.

$$\mathcal{L}_k^{BDKS} = \underbrace{\mathcal{L}_k^{cal}(f_k^{p_W}(x), y) + \alpha \cdot \mathcal{L}_k^{N2W}(f_k^{p_W}(x), f_k^{p_N}(x))}_{\text{training wide BL}} + \underbrace{\mathcal{L}_k^{W2N}(f_k^{p_N}(x), f_k^{p_W}(x))}_{\text{training narrow BL}} \quad (8)$$

where $\alpha > 0$ is the hyper-parameter to control the strength of N2W knowledge distillation, which depends on the model and the dataset. To mitigate the inter-client label skew, we use the loss function of logit calibration $\mathcal{L}_k^{cal}$ here instead of cross-entropy. BDKS can also be extended to multiple groups, as shown in Appendix B.7. The overhead is at the server end which now needs to train $i$ sub-networks for clients in the $i$th group. There is no additional communication and computation overhead on the client end.

## 5 EXPERIMENTAL RESULTS

In this section, we first demonstrate that the proposed *HeteroSFL* achieves significant training-time reduction with minimum accuracy loss for non-IID data. We then study the effectiveness of BDKS in improving the performance of HeteroSFL.

| Time Redn. | 256× | | 128× | | 64× | |
|---|---|---|---|---|---|---|
| **High-end** | **30%** | **60%** | **30%** | **60%** | **30%** | **60%** |
| SR | 17.66 | 29.18 | 35.63 | 40.45 | 42.84 | 55.02 |
| Top-k | 53.97 | 57.43 | 56.60 | 62.65 | 61.01 | 66.60 |
| IdBL | 60.07 | 60.07 | 66.20 | 66.20 | **68.82** | 68.82 |
| HetBL | 53.28 | 60.95 | 56.99 | 61.87 | 60.34 | 63.50 |
| IdBL+dropout | 36.85 | 56.98 | 39.11 | 59.19 | 38.98 | 58.88 |
| HeteroSFL | **61.13** | **65.10** | **67.80** | **69.19** | 68.52 | **69.45** |

Table 1: Accuracy as a function of training time reduction and different proportion of high-end group for $\lambda = 0.05$ data. The data rate ratio between high-end and low-end group is $16 : 1$. Baseline SFL has an accuracy of 70.69% with no reduction in training time.

## 5.1 EXPERIMENTAL SETUP

**Datasets and Models.** We present results for VGG11 on CIFAR-10 in the main paper and ResNet20 on CIFAR-100 in Appendix B.3 due to space limit. For VGG11, the model is split such that the client-side model has 3 convolution layers and for ResNet20 the client-side model has 5 residual blocks. This split is shown to keep the privacy of clients Li et al. (2022b). We provide results for two other layer splitting settings in Appendix B.6. To emulate different degrees of inter-client label skew, the datasets are partitioned with a commonly used Latent Dirichlet Sampling Hsu et al. (2019) with parameter $\lambda$, where lower $\lambda$ represents data with a higher non-IID degree. All experiments are run for 200 epochs. We set the minority class threshold $\theta = 10\%$ (in Section 4.2). The hyper-parameter $\alpha$ for N2W knowledge sharing in Eq.8 changes with models and datasets. The detailed hyper-parameter settings and client-level data distribution are included in Appendix A.

**Heterogeneous clients with different sized BLs.** In this paper, we consider an SFL system with 10 clients and extend it to 50 clients. The clients are separated into two groups – high-end and low-end by default for ease of analysis. We also extend our method to three and four groups. Unless mentioned specifically, we map the clients to groups randomly, which is the setting used in prior works Zhang et al. (2023); Ilhan et al. (2023). The data rate difference between different groups of clients ranges from $4 : 1$ to $16 : 1$, which is reasonable given the study in SpeedTest (2023). The number of channels in the activations generated by the client-side model is 256 for VGG11 and 32 for ResNet20. We sweep the BL size from $1 - 256$ channels for VGG11 and $1 - 32$ for ResNet20 to study the performance of HeteroSFL for different communication time reduction.

**Competing methods** We compare the performance of HeteroSFL with baseline SFL Thapa et al. (2022) and several other competing methods. **SampleReduce (SR)** reduces the training time by reducing the number of samples processed by clients per epoch. **Top-k sparsity** Liu et al. (2018) reduces the training time by sending the activations with top-k magnitude to server. **IdBL** Ayad et al. (2021) uses identical-sized BLs in both high- and low-end groups. We also compare HeteroSFL with a variant of IdBL method, **IdBL+channel-wise Dropout (IdBL+dropout)** Hou & Wang (2019), where the high-end and low-end clients use identical-sized BL, and the client picks up $p$ channels based on the magnitude of activations of different channels and zeroes out the other channels in the BL in every epoch. We implement Logit Calibration Zhang et al. (2022) for all competing methods, except baseline SFL.

## 5.2 TRAINING-TIME AND ACCURACY PERFORMANCE OF HETEROSFL

**HeteroSFL achieves the best accuracy compared with competing methods.** Table 1 compares the accuracy of HeteroSFL with all other competing methods as a function of training time reduction and proportion of clients in high-end group for $\lambda = 0.05$ data. The ratio of data rates between high-end and low-end group of clients is $16 : 1$. Thus to reduce time by $256\times$, the communication data of low-end group is reduced by $256\times$, and high-end group by $16\times$. Given the same training time reduction and proportion of clients in high-end group, SampleReduce and Top-k sparsity achieves $30.07\%$ and $7.16\%$ lower accuracy than HeteroSFL on average, respectively. The better performance of HeteroSFL indicates that BL can compress the communication data by preserving more useful information. For IdBL, both high-end and low-end group are compressed by the same ratios, therefore the proportion of high-end group does not impact its performance. When the training time

Table 2: Accuracy of HeteroSFL with three and four groups for $\lambda = 0.05$ data. The training time reduction is $256\times$. The data rate ratio for different groups are $1 : 8 : 16$ for three groups and $1 : 4 : 8 : 16$ for four groups.

| | Proportion of clients | SR | Top-k Sparsity | IdBL | HetBL | HeteroSFL |
|---|---|---|---|---|---|---|
| **Three Groups** | **60%, 20%, 20%** | 20.31 | 53.69 | 60.07 | 49.09 | **60.75** |
| | **20%, 40%, 40%** | 22.90 | 58.72 | 60.07 | 61.66 | **64.08** |
| | **20%, 20%, 60%** | 30.19 | 60.50 | 60.07 | 63.47 | **66.79** |
| **Four Groups** | **30%, 30%, 20%, 20%** | 29.14 | 57.26 | **60.07** | 55.12 | 59.24 |
| | **20%, 20%, 20%, 40%** | 24.98 | 60.16 | 60.07 | 60.08 | **61.52** |
| | **10%, 10%, 20%, 60%** | 32.99 | 61.26 | 60.07 | 62.55 | **64.59** |

reduction is $256\times$, HeteroSFL outperforms IdBL by $5.03\%$ and $1.08\%$ when the number of clients in high-end group is 60% and 30%, respectively. HeteroSFL has better performance due to more information sent by the high-end group of clients.

**Higher training time reduction results in higher accuracy loss.** Baseline SFL has $70.69\%$ accuracy with no training time reduction. Compared to SFL, all competing methods achieve significant training time reduction with accuracy loss, with HeteroSFL achieving the lowest accuracy loss for the same training time reduction. HeteroSFL reduces training time by $64 \times -256\times$ at the cost of $2.17\% - 9.56\%$ accuracy loss when 30% of clients are in high-end group. When the number of clients in high-end group increases to 60%, the accuracy loss reduces to $1.24\% - 5.59\%$, since more information is sent to the server by the high-end group.

**Different degrees of inter-client label skew.** Table 4 and Table 5 in Appendix B.1 show the performance of HeteroSFL for $\lambda = 0.3$ and IID data. Compared with baseline SFL, HeteroSFL reduces training time by $16\times$ to $128\times$ with $0.28\%$ to $3.54\%$ accuracy loss for $\lambda = 0.3$ non-IID data and $1.46\%$ to $4.14\%$ accuracy loss for IID data. Given the same training time reduction, HeteroSFL outperforms SampleReduce, Top-k sparsity, and IdBL by up to $35.59\%$, $14.35\%$ and $6.98\%$ for $\lambda = 0.3$ data, and $25.63\%$, $5.93\%$ and $9.90\%$ for IID data.

**Different data rates.** Table 6 in Appendix B.2 shows the performance of HeteroSFL when the data rates between high-end and low-end group change from $4 : 1$ to $16 : 1$ for $\lambda = 0.05$ data. Given $256\times$ training time reduction, HeteroSFL outperforms SampleReduce, Top-k sparsity, and identical BL by up to $49.47\%$, $12.46\%$ and $4.07\%$, for $4 : 1$ data rate ratio, and $34.98\%$, $4.70\%$ and $5.03\%$ for $16 : 1$ data rate ratio. The improvement of HeteroSFL over IdBL becomes higher when the data rate difference is larger since the additional information sent by high-end group increases.

**Increasing number groups** Table 2 shows the performance of HeteroSFL when there are three and four groups of clients for $\lambda = 0.05$ data. With $256\times$ training time reduction, HeteroSFL achieves up to $41.48\%$, $7.06\%$ and $6.72\%$ higher accuracy compared with SampleReduce, Top-k Sparsity, and IdBL.

**Performance for ResNet20 on CIFAR100** We also compare the performance of HeteroSFL with other competing methods for ResNet20 on CIFAR100 for different data distributions. The results are shown in Appendix B.3 due to space limitations. Given the same training time reduction, HeteroSFL outperforms SampleReduce, Top-k sparsity, and IdBL by up to $10.48\%$, $5.69\%$ and $7.48\%$ for $\lambda = 0.05$ data, and $9.53\%$, $7.71\%$ and $7.58\%$ for $\lambda = 0.3$ data.

## 5.3 Effectiveness of BDKS in addressing inter-group label skew

Table 3 shows the performance of HeteroSFL and HetBL for 50 clients when the data label distribution and client heterogeneity are correlated as in Maeng et al. (2022) and when they are uncorrelated. For the case when there is no correlation between the two types of heterogeneity, we map the clients to groups randomly. Even in this case there is a low level of inter-group label skew due to the severe inter-client label skew and BDKS helps improve accuracy by $3.77\%$ over HetBL when the high-end group has 30% clients. To generate data where the data label distribution and client heterogeneity are correlated , we keep the inter-client label skew as $\lambda = 0.05$, and map the clients with similar

Table 3: Accuracy of HetBL and HeteroSFL with 50 clients. When the training time reduction is $256\times$ and the data rate ratio between high-end and low-end group is $16:1$.

| # of clients | High-end group | HetBL | HeteroSFL |
|---|---|---|---|
| **50-clients** | 30% | 57.15 | **60.92** (+3.77) |
| | 60% | 61.43 | **63.34** (+1.91) |
| **50-clients** | 30% | 53.22 | **59.16** (+5.94) |
| **(Correlated)** | 60% | 61.38 | **64.35** (+2.98) |

label distribution to the same group. Such a case results in a higher level of inter-group label skew, and BDKS improves accuracy even more by $5.94\%$, as shown in Table.3.

The inter-group label skew caused by severe inter-client label skew is even more significant with small number of clients. Simply using HetBL in 10-client SFL system results in lower accuracy than IdBL (see Table 1) due to the presence of inter-group label skew. For $256\times$ training time reduction, BDKS improves the accuracy over HetBL by $4.15\% - 7.85\%$ and helps HeteroSFL beat IdBL. IdBL+dropout achieves $18.39\%$ lower accuracy on average compared with HeteroSFL and still performs worse than IdBL, indicating that training different channels alternately fails to address the inter-group label skew. Fig.3 shows the class-wise accuracy of HeteroSFL with and without BDKS for $40\%$ of high-end group in 10-client SFL system. The accuracy of underrepresented classes (i.e. classes 5 to 9) increases significantly, indicating BDKS

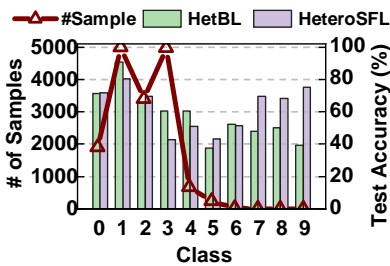

Figure 3: Class-wise accuracy of HetBL and HeteroSFL with 40% high-end group for 10 clients.

alleviates the overfitting of the wide BL model. We also evaluate the performance of BDKS for ResNet20 in a 10-client system and the detailed results are shown in Table 9 Appendix B.4. We see that BDKS helps improve accuracy by up to $6.30\%$.

**Wide-to-Narrow Knowledge Sharing** We replace W2N knowledge sharing in BDKS (Eq.8) with logit calibration loss, referred to as HeteroSFL w/o N2W; the corresponding results are shown in Table 10 in Appendix B.5. For ResNet20, the accuracy of HeteroSFL w/o N2W is significantly worse than HeteroSFL, especially with $80\%$ clients in high-end group. This illustrates that W2N knowledge sharing is necessary to train the narrow BLs in high-end group for higher accuracy.

## 6 CONCLUSION AND FUTURE WORK

In this paper, we propose *HeteroSFL*, the first SFL framework with heterogeneous clients that process non-IID data characterized by inter-client and inter-group label skew. HeteroSFL utilizes HetBL to address the different compression ratios of different groups through different-sized BL, logit calibration Zhang et al. (2022) to mitigate the inter-client label skew, and BDKS to address the inter-group label skew. For a 10-client SFL system, HeteroSFL can reduce the training time of SFL by $16\times$ to $256\times$ with $1.24\%$ to $5.07\%$ accuracy loss, on average, for VGG11 on CIFAR10 data with significant label distribution skew. While this work tackled handling non-IID data with significant label distribution skew in the heterogeneous client setting, in future we plan to address other types of inter-group data distribution differences such as those with domain shift. We also plan to study fairness issues in such heterogeneous SFL settings.

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
