# OpenReview forum: "HeteroSFL: Split Federated Learning with heterogeneous clients and non-IID data"
_ICLR.cc/2024/Conference — ICLR 2024 Conference Withdrawn Submission_

### Official Review · Reviewer_NpV1 · 2023-10-26

**Soundness:** 2 fair
**Presentation:** 1 poor
**Contribution:** 2 fair
**Rating:** 5
**Confidence:** 4

**Summary:**

This paper investigates split federated learning (SFL) and proposes the HeteroSFL framework, which can reduce the training time and tackle the non-IID problem. This is an interesting work in the field of SFL.

**Strengths:**

As claimed by the author, the proposed HeteroSFL method is the first SFL framework for heterogeneous clients. It can tackle inter-client and inter-group label distribution skew while reducing the training time. This has been verified by extensive experimental results.

**Weaknesses:**

When comparing the training time, the federated learning methods (e.g., FedAvg) should be used as a baseline in the experiment. The main motivation for developing SFL methods is that the classical FL training schemes need to deploy the entire model on clients, which leads to high computation latency. However, the drawback of SFL is that the clients need to transmit the activations and receive the gradients in each training round, which may result in a much higher communication overhead than that of the FL methods. Besides, most of the FL methods (e.g., FedAvg) can iterate the local update multiple times before aggregation. This reduces the communication frequency and thus further reduces the training time. Given the above, the reviewer supposes that low upload or download rates will form a bottleneck for the SFL methods.

**Questions:**

1. Revise typos in the manuscript. For instance, "pickss" should be "picks" in "the client pickss up p channels ..."

2. The authors should check the format of the references.

---

### Official Review · Reviewer_iigy · 2023-10-28

**Soundness:** 2 fair
**Presentation:** 2 fair
**Contribution:** 3 good
**Rating:** 6
**Confidence:** 4

**Summary:**

Split Federated Learning (SFL) is a privacy-preserving decentralized learning scheme that splits a machine learning model to offload most computations to the server. SFL suffers from high communication costs and heterogeneity problem in practical scenarios. This paper focuses on the heterogeneity problem of SFL which is of great importance. They propose HeteroSFL that handles non-IID data with label distribution skew across clients and across groups of clients. HeteroSFL compresses data with different compression factors in low-end and high-end groups using narrow and wide bottleneck layers (BL), respectively. It provides a mechanism to aggregate different-sized BL models and utilizes bidirectional knowledge sharing (BDKS) to address overfitting caused by different label distributions across high- and low-end groups. Experimental results show that HeteroSFL achieves significant training time reduction with minimum accuracy loss compared to competing methods.

**Strengths:**

1.	The heterogeneity problem in SFL is of great importance in practical settings. This paper greatly mitigates the issue.
2.	The idea of bidirectional knowledge sharing is novel to some extent.
3.	The experimental results are promising which demonstrates superior performance than baselines.

**Weaknesses:**

1.	There are several important baselines that this paper should compare with. Since this paper says that they focus on compressing the communication and tackling the NonIID data, communication compression strategies in SFL should naturally be compared. The typical works include [a].
[a] Jianyu Wang, Hang Qi, Ankit Singh Rawat, Sashank Reddi, Sagar Waghmare, Felix X Yu, and GauriJoshi. Fedlite: A scalable approach for federated learning on resource-constrained clients. preprint arXiv:2201.11865, 2022

2.	The theoretical convergence is not guaranteed.

**Questions:**

1.	In the third paragraph, this paper says that “Since the client-side computation in SFL is quite small, we do not consider it. Instead, we focus on reducing the communication cost by compressing the transmitted/received data using BL”. However, in Section 4.1, this paper applies different bottleneck layers for different groups, which is similar to HeteroFL. This confuses me whether this paper consider this problem.

2.	The idea of bidirectional knowledge sharing is similar to ensemble knowledge distillation for federated learning [b]. Will it be better to direct apply an ensemble distillation over all clients or groups instead of using bidirectional knowledge sharing?
[b] Tao Lin, Lingjing Kong, Sebastian U. Stich, Martin Jaggi: Ensemble Distillation for Robust Model Fusion in Federated Learning. NeurIPS 2020

3.	Where does the data come from for bidirectional knowledge sharing?

---

### Official Review · Reviewer_wycu · 2023-11-03

**Soundness:** 3 good
**Presentation:** 3 good
**Contribution:** 2 fair
**Rating:** 3
**Confidence:** 4

**Summary:**

This paper proposes a SFL framework under heterogeneous clients context. The idea is to divide the bottleneck layers to different size to accomodate different communication ability of clients. To futher accomodate the performance loss due to employing various bottleneck size, the authors futher prooses directional knowledge sharing for exchacing knowledge between different sub-network.

**Strengths:**

1. The paper is well-written and easy to read. The integration of different techniques has clear motivation.

2. The experiment results demonstrate good performance.

**Weaknesses:**

1. The novelty of this paper is not sufficient. The paper seems to be a bundle of multiple existing techniques. To be specific, the authors integrate bottleneck technique by bottleneck compression by Eshratifar et al. (2019); Ayad et al. (2021),  logit calibration by (Zhang et al, 2022), subnetwork aggregation in HeteroFL (Diao et al, 2021), knowledge distillation between sub-network by (Ilhan et al, 2023). Given this, the contribution seems to be rather incremental as all of these techniques have been applied in SFL or FL setting.  The only novel contribution is bi-directional knowledge transfer improving upon single direction knowledge transfer proposed by (Ilhan et al, 2023). However, the bi-directional knowledge distillation (or mutual learning) has also been explored in federated learning before (Shen et al. 2020).

Shen T, Zhang J, Jia X, et al. Federated mutual learning[J]. arXiv preprint arXiv:2006.16765, 2020.

2. The experiment is not comprehensive.
    - The authors only compare their methods with three baselines, SR  and TOP-K sparsity, and IdBL. However, the authors do not compare other methods with those baselines that consider heterogeneous training settings, e.g., ScaleFL (Ilhan et al, 2023), and HeteroFL (Diao et al, 2021). Note that these methods are model-agnostic, which means they can be easily applied to IdBL with bottleneck layers in the model.
To ensure fair competition, I suggest the authors to compare those two methods in two groups of experiment,  with the first group assuming the presence of logit calibration,  and the second group without it, as both of these two methods do not apply this technique in their vanilla version.
   - The experiment is done only in small datasets CIFAR10, and CIFAR100.
   - The authors did not do an ablation study. The methods bundle a lot of components. It is essentially to see the impact of each component.
   - The baselines in CIFAR10 task do not reach comparable accuracy, which at least reaches 80% even in Non-IID setting.
   -  The authors should compare their methods under different Non-IID settings. Please consider doing experiment setting $\lambda= 0.5$, $\lambda= 1$, and IID setting.


4.  Privacy concern.
    - The BL method needs to send the compressed representation to the server, which poses serious threats of data leakage. As FL is mostly concerned with privacy protection, sending activation alone is too risky.
    - (major) The clients in the high-end group need to be aware of R, i.e., the set of underrepresented classes. However, in the FL setting,  knowing the statistical distribution of data of others alone leaks privacy.

5. Minor: the paper itself is not self-contained. The authors integrate logit calibration by (Zhang et al, 2022), but do not show how it calibrates the label heterogeneity.

**Questions:**

1. How do the authors design the baseline method DiffBL? The authors did not descibe technical details of this baseline method, and this baseline is not proposed by any literature. My concern is that the baseline might not be reliable and therefore the peformance gain of HetBL  may not be objective.

2. Why the second term in (6) ensures that the wide BL model does not learn about J classes from the narrow BL model, while the first term  helps the wide BL model to learn about underrepresented R classe? The second term differs from the first term only in that the sum is in different positions.

3. Concern of communication reduction. Is every client getting the same model? Can we achieve communication reduction by only sending the corresponding BL used by each client?

4. How do you simulate label skewness between groups of clients (as in Figure 2 (a))?